# TiCoCrFeMn (BCC + C14) High-Entropy Alloy Multiphase Structure Analysis Based on the Theory of Molecular Orbitals

**DOI:** 10.3390/ma14185285

**Published:** 2021-09-14

**Authors:** Dominika Gorniewicz, Hubert Przygucki, Mateusz Kopec, Krzysztof Karczewski, Stanisław Jóźwiak

**Affiliations:** 1Faculty of Advanced Technologies and Chemistry, Military University of Technology, Sylwestra Kaliskiego 2, 00-908 Warsaw, Poland; krzysztof.karczewski@wat.edu.pl (K.K.); stanislaw.jozwiak@wat.edu.pl (S.J.); 2GeniCore Sp. z o.o., Wolczynska 133, 01-919 Warsaw, Poland; hubert.przygucki@genicore.pl; 3Institute of Fundamental Technological Research, Polish Academy of Sciences, Pawińskiego 5B, 02-106 Warsaw, Poland; mkopec@ippt.pan.pl or; 4Department of Mechanical Engineering, Imperial College London, London SW7 2AZ, UK

**Keywords:** HEA, solid solution, laves phase, U-FAST sintering

## Abstract

High-entropy alloys (HEA) are a group of modern, perspective materials that have been intensively developed in recent years due to their superior properties and potential applications in many fields. The complexity of their chemical composition and the further interactions of main elements significantly inhibit the prediction of phases that may form during material processing. Thus, at the design stage of HEA fabrication, the molecular orbitals theory was proposed. In this method, the connection of the average strength of covalent bonding between the alloying elements (Bo parameter) and the average energy level of the d-orbital (parameter Md) enables for a preliminary assessment of the phase structure and the type of lattice for individual components in the formed alloy. The designed TiCoCrFeMn alloy was produced by the powder metallurgy method, preceded by mechanical alloying of the initial elementary powders and at the temperature of 1050 °C for 60 s. An ultra-fine-grained structured alloy was homogenized at 1000 °C for 1000 h. The X-ray diffraction and scanning electron microscopy analysis confirmed the correctness of the methodology proposed as the assumed phase structure consisted of the body-centered cubic (BCC) solid solution and the C14 Laves phase was obtained.

## 1. Introduction

High-entropy alloys (HEA) are currently enjoying great interest as evidenced by the increasing number of publications on these materials. The reason for such progressively developing research in this area is the practically unlimited variations of the chemical compositions that enables for the control of the expected functional properties, for example, high strength under both cryogenic and high-temperature conditions [1]. According to the literature, high-entropy alloys are based on at least five alloying elements (5–35%), with a single-phase matrix that consists of solid solutions: regularly face-centered cubic (FCC) or body-centered cubic (BCC) [2,3]. In recent years, the concept of high-entropy alloys has also been expanded to multicomponent and multiphase materials [4,5,6,7,8,9,10,11], which in their phase structure, apart from solid solutions, also contain intermetallic compounds or even amorphous phases with high-entropic properties [12]. This approach significantly extends the unique properties of HEA and their potential applications. Materials with the participation of the Laves phase, as the second component of the phase structure, are of a particular interest among multi-phase HEA. These specific AB_2_-structured materials include a large group of intermetallic compounds with many interesting properties in the field of superconductivity [13,14,15], magnetism, magnetocaloric [16] and hydrogen absorption [17,18,19]. Titanium-doped HEA are also used in specialized and extreme applications in Generation IV nuclear reactors [20]. The reason for significantly lower interest in high-entropy multiphase alloys is probably associated with the complexity of the design processes for the assumed phase structure related to the amount and type of component elements used for the material fabrication.

The classic design approach of HEA is based on the phase structure assessment using thermodynamic parameters such as: mixing entropy (ΔS_mix_) at the level of 1.6R (R-gas constant), mixing enthalpy ΔH_mix_, metallic radius size coefficient σ, parameter determining the melting temperature dependence Ω and the resultant concentration of the valence electrons VEC (valance electron concentration). It should be mentioned, however, that this approach only considers the presence of the FCC/BCC crystal lattice or their mixtures in the volume of the designed material. By controlling the above-mentioned thermodynamic parameters, it is only possible to obtain maps showing the formation of a solid solution, an intermetallic compound, amorphous phases and metallic glasses. Depending on the composition of the high-entropy alloy, it would be possible to determine the presence of a given phase in this alloy. The main difficulty occurs when a given alloy is multiphase or the obtained point has an ambiguous position in the graph of the mentioned thermodynamic relationships. Therefore, one should consider alternative methods for the final phase structure prediction in terms of designing new, multiphase alloys with high-entropic features. Zhang et al. [21] and Kozak et al. [22] presented an approach for phase structure assessment using the traditional trial and error method. However, such an approach is time-consuming and cost-intensive because of the number of potential chemical composition combinations. Another approach presented in the literature is the Calculation of Phase Diagrams (CALPHAD) method, which is extremely helpful in areas related to the modeling of phase transformations. The CALPHAD method enables for the calculation of higher order systems considering many phases in investigated materials [21]. However, this method requires access to thermodynamic databases. 

An alternative method to the conventional approaches, namely the theory of molecular orbitals, was proposed to predict the assumed phase structure of the alloy. According to this method, the connection of the average covalent bond strength between the alloying elements (Bo parameter) with the average energy level of the d-orbital (parameter Md) enables for the preliminary estimation of the phase structure and the lattice type for individual components [23,24,25,26,27] based on the analysis of two-component equilibrium systems of the elements comprising the designed alloy.

This method, patented in 1987 by Yukawa et al. [28], was described in [26] on the example of δ ferrite formation in ferritic steels, which also consider hydrogen absorption and desorption in metal alloys. The correctness of this method for the analysis of the martensitic transformation in titanium alloys was also confirmed in numerous studies analyzing phase transformations in conventional titanium alloys [24,29,30,31]. This concept was also used for the analysis of the phase transformation-induced plasticity (TRIP) occurring in high-entropy alloys with dominant titanium content [32,33,34]. It should be noted, however, that these works concern the changes that occurred in the conventional alloys with a solid solution structure without the precipitation of other phases, such as the Laves phase. The problem of solubility limit analysis of the alloying elements in high-entropic solid solutions was analyzed in [35], where the limit value of the Md parameter was determined for BCC solutions based on iron and chromium (Md_Fe_ = 0.92 and Md_Cr_ = 0.86, respectively). The authors of this study also suggest that these values could also be used in high-entropy alloys with a dominant content of titanium, and additional iron and/or chromium in the alloy composition.

In order to confirm the potential application of the theory of molecular orbitals for the design of multicomponent two-phase, high-entropy alloys with the assumed phase structure and participation of the Laves phase, the chemical composition of the historically first, comprehensively tested Cantor alloy CoCrFeMnNi [36] was modified. Otto et al. [37] found that the titanium-based alloy doped with Cr, Fe, Mn and Ni was characterized by a desirable, multiphase structure. Moreover, the combination of Ti, Cr and Mn formed Laves phases with a BCC solid solution and the addition of titanium led to the formation of both a Laves phase and FCC crystal lattice stabilized by nickel [38,39]. The stabilization of the FCC lattice by nickel was also confirmed by X-ray diffraction of the chromium-free alloy Co_25_Fe_25_Mn_5_Ni_25_Ti_20_, where the coexistence in the structure of BCC and FCC solutions as well as the Laves Ti_2_ (Co, Ni) phase was identified [40]. The ability of the titanium to form a Laves phase was also described by Liu et al. [41], who investigate a wear-resistant, lattice FCC × phase and Laves phase-based CoCrFeNiTi coatings. The performed literature analysis indicates that the nickel addition ensures the stabilization of the FCC solid solution lattice, and on the other hand, the presence of titanium led to the formation of the Laves phase. Therefore, in order to obtain the structure of the BCC and the intermetallic Laves phase in the designed alloy, a TiCoCrFeMn alloy, not yet investigated in the literature, was proposed. In such an alloy, nickel was replaced by titanium due to the following reasons:Titanium has a hexagonally compact crystal lattice at room temperature, which is characterized by higher strength properties than nickel, at a much lower density;Titanium easily forms a BCC-structured solid solution with elements such as V, Cr, Fe, Mo, Cu, Nb, Ta and Mn;The melting point of titanium is about 200 °C higher than that of nickel (1660 °C to 1453 °C);Titanium has paramagnetic properties, which extends its possible functional applications;Toxic nickel has harmful effects on human health, while titanium remains biocompatible.

It should be highlighted, that the ease of Laves phases formation with the C14 structure by titanium, mainly in the presence of iron, chromium and manganese, determined the choice of this element for the fabrication of the novel HEA. The high entropy of mixing resulted in the formation of a solid solution, which did not compensate for the negative effect of the mixing enthalpy, and thus led to the formation of an intermetallic structure [42]. The Laves phase with the C14 structure is one of the most common intermetallic phases. At the same time, due to the aforementioned properties, it has great application potential.

Thus, this paper presents the development of TiCoCrFeMn alloy with a specifically selected chemical composition that has not yet been investigated in the literature. It was assumed that the designed alloy will have a two-phase structure consisting of a BCC solid solution and a Laves phase, and such structure will be verified by the classical material characterization methods (scanning electron microscope (SEM), X-ray diffraction (XRD)) and the proposed Bo–Md diagram. Moreover, the effectiveness of such methodology to predict the phase composition of different alloys before they will be manufactured will be investigated as well.

## 2. Materials and Methods

### 2.1. Experimental Methods

The powder metallurgy method was used to manufacture the two-phase HEA. This method enables precise control of the chemical composition of the produced material. The mixture of Ti, Co, Cr, Fe and Mn initial powders (Table 1), with an average particle diameter size under 45 μm, was mechanically synthesized for 5 h in a vacuum using a planetary ball mill Fritsch Pulverisette 7 Premium (Fritsch GmbH, Idar-Oberstein, Germany) and grinding the media with a diameter of 10 mm made of 100Cr6 bearing steel in an 8: 1 ratio.

The synthesized powder mixture was sintered by using a modified spark plasma sintering (SPS)-based method, namely U-FAST. This method enables an application of an ultrafast heating rate and subsequent sintering of a fine-grained structure, which is desirable in accordance with the Hall–Petch relationship to improve the strength properties of the formed material [43]. The compacted material was subjected to short, microsecond current pulses of high intensity at ~2000 A with a pressure force of 13 kN, using a heating rate of 50 °C/min to consolidate the powder mixture. The sintering process was carried out at the temperature of 1050 °C for one minute. Then, the sinters of 20 mm diameter and 10 mm height were annealed under an argon protective atmosphere in the PRC 50 × 470/110M tube furnace (Czylok, Jastrzębie-Zdrój, Poland) at a temperature of 1000 °C for 1, 10, 20, 50, 100 and 1000 h to assess the structural stability of the phase structure obtained. Metallographic specimens were firstly ground on SiC sandpaper of 600 ÷ 4000 grade, and then polished using cloths and a diamond suspension of 3 µm and 0.25 µm and a silica suspension of 0.1 µm. The alloy structures were observed on the Quanta 3D FEG field emission scanning electron microscope (SEM) (Field Electron and Ion Company, FEI, Hillsboro, OR, USA), which allowed us first to perform a chemical composition analysis using an energy-dispersive X-ray spectroscopy (EDS) detector and an additional microdiffraction using an electron backscatter diffraction detector (EBSD). The phase fractions in the material were determined by using X-ray diffraction (XRD) analysis on a Rigaku X-Ray Diffractometer Ultima IV (Rigaku, Tokyo, Japan) with a Co lamp (λ = 1.79 Å) and PDF-4 database. The XRD measurements were repeated on a Seifert XRD 3003 TT diffractometer (Seifert, Mannheinm, Germany) with a Cu lamp (λ = 1.54 Å) to identify plane reflections due to the different absorption of radiation of the elements. Based on the obtained diffractograms, the specific crystal lattices were estimated, cubic lattice parameters of the BCC solid solution were calculated and the parameters of the hexagonal C14 lattice by the Cohen’s method were obtained.

### 2.2. Design of High-Entropy Alloys

The chemical composition of the designed alloy was selected to develop a phase system consisting of a BCC solid solution and a C14 intermetallic Laves phase with a hexagonal lattice. The assumed composition of TiCoCrFeMn meets the criterion described in [44] for materials intended for application in IV generation fission and fusion reactors. This criterion assumes that the radioactivity of materials used in extreme conditions declines to a low level of activity after further irradiation. Based on the British standards, only elements of low-level waste that will degrade within 100 years could be used in the construction of nuclear power components. In the case of the investigated TiCoCrFeMn alloy, it could be concluded that this condition has been met. The time of degradation for Ti and Cr is 10–50 years, for Mn and Fe it is 50–100 years and for Co it is 100–300 years [44]. Such knowledge extends the possible applications for the material in question to the nuclear energy sector.

As the composition of the high-entropy alloy usually consists of at least five different elements, its design cannot be based on conventional equilibrium systems. It was found, however, that for Ni [25], Fe [26] and Ti [27] based alloys, the molecular orbitals method can be successfully used to predict and determine their phase structure. The molecular orbital method, called Bo–Md, consists of calculations of two parameters for the main element in an alloy. The Bo parameter (visualization on Figure 1a) is short for bond order and describes the strength of bonding between the main element and the alloying element. The bond order value differs in the case of different crystal structures, as shown on Figure 2. The alloying element has a different number of neighbors in a different distance in the cubic, body-centered lattice (Figure 2a) and is different in the hexagonal close-packed lattice (Figure 2b) Md parameter (Figure 1b), which is short for metal d-orbital energy and connects the electronegativity and atomic radius of the alloying elements together with the main element in an alloy. The energy level in the hybridized bond between the main and alloying element controls the charge transfer direction, so it is related to electronegativity. A higher main element d-level energy (Figure 1b) means that the charge transfer is from the main element to the alloying element. The Md parameter is influenced by the atomic radius [45], so the radius of the d-orbital is increased, and thus its energy level is higher [25]. For calculating both of the parameters for the complex alloy, the formulas shown below are used:(1)$$\bar{Md}=\overset{n}{\underset{i=l}{\sum }}X_{i}{\left(Md\right)}_{i}$$
(2)$$\bar{Bo}=\overset{n}{\underset{i=l}{\sum }}X_{i}{\left(Bo\right)}_{i}$$
where, *X_i_* is the atomic fraction of component *i* in the alloy, and (*Md*)*_i_* and (*Bo*)*_i_* are the values for *i* component.

Therefore, on the basis of the methodology presented in [32], a completely new Bo–Md diagram, presented in Figure 3, was developed for the purposes of the presented research. The positions of the individual elements on the system axes were determined using the literature data [27]. Subsequently, the points corresponding to the individual elements were connected with lines. Each line connecting these two elements corresponded to a double equilibrium system. These lines indicate the areas of individual phases occurrence characterized by a specific crystal lattice. The analysis of Figure 3a enables us to conclude that the dominant phases in the diagram show the structures of the BCC, FCC and hexagonal one. Therefore, in order to illustrate the main phase components of an alloy with any assumed chemical composition, areas with a different crystal structure were shown as gray zones (Figure 3b). The distribution map of these crystallographic lattices allows us to predict the phase structure for a larger number of alloys. The combination of areas for the same types of lattices determined for all of the pairs of elements presented allowed for the creation of a Bo–Md diagram, on which, taking into account the chemical composition of the designed alloy, the material in question could be located. Based on the results of the proposed methodology, it could be concluded that the investigated TiCoCrFeMn alloy is located on the border of the BCC and hexagonal lattice. Such location suggests that the final phase structure will consist of the BCC solid solution and the hexagonal lattice identified with the C14 structure of the Laves phase. Additionally, the obtained experimental results were compared with the compositions from the literature [3], where the CoCrFeMnNi alloy is located in the FCC lattice area and the NbHfTaTiZr alloy is in the BCC lattice area, and these data are consistent.

## 3. Results and Discussion

Band contrast maps (BC) made with the EBSD technique enable us to visualize areas of low contrast, such as the grain boundaries and interfacial boundaries. They allowed us to characterize the microstructure of the material in question as an ultra-fine grained one (Figure 4a,c,e). The sinter matrix consists of grains smaller than 1 µm and local areas of grain size of about 5 µm. These local areas dissolve with the annealing time, rebuilding into a fine-grained structure after a long homogenization period of 1000 h. On the other hand, the observations performed by using the backscattered electron (BSE) detector (Figure 4b,d,f) were found as sensitive to changes in the chemical composition of individual phase components. Such an observation revealed that the grain structure of the alloy consists of grains significantly different in contrast, which proves that the assumed two-phase structure was obtained. Therefore, taking into account the grain size and differences in contrast, and in order to clearly describe the analyzed areas at this stage of work, the following nomenclature was used:Small bright area—sb—for areas with a bright matrix phase contrast;Small gray area—sg—for areas of the dark gray matrix phase contrast;Large bright area—lb—for areas with bright phase contrast and a grain size of about 5 µm.

Despite the protective atmosphere during the technological process, the presence of numerous oxide precipitates evenly distributed in the material matrix and coagulating during the annealing treatment was found in the sintered structure. It was concluded that the formation of these ceramic precipitates was caused by the leakage of the system that occurred during the grinding process. Subsequent contact of the reactive powders with oxygen resulted in the formation of titanium oxides in the alloy structure.

The microanalysis of the chemical composition in the individual phase regions showed that the bright regions, both the ultra-fine-grained in the two-phase zone and in the local single-phase microregions, mainly consist of elements with a crystalline BCC lattice-based structure (Cr, Fe, Mn). On the other hand, in the grey areas, a noticeable increase in the content of hexagonal lattice-based elements, i.e., titanium and cobalt were found (Figure 5). This observation suggests that, in accordance with the theory of molecular orbitals modeling (Figure 3), the assumed two-phase sinter structure that was based on the BCC lattice of the solid solution and the hexagonal lattice of the Laves phase was obtained. Nevertheless, in order to identify the crystallographic structure of the observed areas with the different phase contrast, EBSD microdiffraction studies were performed in the areas of “large bright particles” on the basis of the crystal lattice parameters values obtained from the XRD tests. The cross in Figure 6a indicates the area of the analysis in which the electron diffraction was recorded (Figure 6b). The Kikuchi lines shown in Figure 6b were indexed according to the declared lattice parameters. Additionally, red dots indicate the sites subjected to the analysis with the same parameters as these point with a cross. The obtained Kikuchi line system clearly indicates that the bright phase has the structure of a BCC with a lattice constantly oscillating around the value characteristic for the manganese lattice.

The identified phase regions that differ in contrast also significantly differ in their chemical composition. The bright phase is based on an increased chromium content, while the grey phase areas are characterized by an increased proportion of titanium and cobalt (Table 2). The analysis of the chemical composition in individual phase regions after homogenization at 1000 °C revealed that the grey phase regions are mainly depleted in chromium, while the titanium content increases (Figure 7). In the case of the bright phase, the diffusion flux of these two elements is opposite. The analysis of the chemical composition enables us to conclude that the bright phase is mostly composed of elements that nominally have a cubic lattice. On the contrary, the grey ones are dominated by Ti and Co, which at room temperature have a hexagonal lattice. Such observations combined with the results considering the values of the enthalpy for individual atomic pairs formation presented in [46] allowed us to state that the gray phase regions, as the AB_2_ phase with a hexagonal lattice, were mainly formed from the reactions between titanium (cobalt) and the remaining alloy components. On the other hand, the formation of the bright areas identified as the BCC solid solution was mainly related to chromium fluctuations.

The analysis of the chemical composition evolution has shown that the significant change in the proportion of two basic elements affecting the phase structure of the alloy occurred after 10 h of homogenization at 1000 °C. Such a phenomenon was attributed to the slow diffusion effect, typical for HEA, which is responsible for the high thermal stability of multicomponent high-entropy alloys [3]. An additional confirmation of the largest fluctuations in the chemical composition at this annealing time interval was also shown during the analysis of changes in the standard deviation of the content of individual elements (Figure 8). As could be observed, the general decreasing tendency of the standard deviation in the proportion of Ti and Cr in the two-phase fine-grained matrix volume indicates the stabilization of the chemical composition. Thus, the phase stabilization was effectively disturbed for the homogenization time of about 10 h. This effect, combined with changes in the content of individual elements (Figure 7), indicates that intense diffusion processes leading to the formation of the final structure occurred at a temperature of 1000 °C in the range from 1 to 20 h of annealing.

Changes in the chemical composition observed by using microscopic methods could be also observed on Bo–Md diagram (Figure 9). The calculations of the covalent bond average strength (Bo parameter) and the average d-orbital energy level (Md parameter) for the actual content of the elements (Table 1) revealed locations of characteristic points. Those points and their relative movement on the Bo–Md diagram in function of the annealing time shows that the “grey” phase is moving towards the hexagonal region and the “small bright phase” is going in direction of the BCC regions. Moreover, it could be observed, that the diffusion that occurred in large bright regions leads to changes in the chemical composition. Those changes result in the subsequent formation of a solid solution with a BCC structure characterized by the same parameters as for an ultra-fine-grained matrix. Such behavior was associated to the homogenization in the regions of solid solution formed during sintering process. This homogenization was performed at a temperature of 1000 °C and time equal to 10 h. Such a long time required for incubation could be a proof of a sluggish diffusion, characteristic for HEA. Thus, by using the methodology proposed in the work of [35], it was assumed that in the investigated alloy TiCoCrFeMn, the border value Md_Ti_ for the parameters Ti-M [28], below which a solid solution with a BCC structure occurred, equals to 1.375. On the other hand, the Laves phase with a C14 structure are forming above 1.40 values of Md_Ti_. The analysis Bo–Md diagrams from [29,30,31,32,33,34] allows us to state that a Ti-based BCC structure occurs with Md values much higher, even above 2.5, which is evidently dependent of Bo values. Sheikh et al. [35] suggest that border the Md parameter that characterizes precipitations in Fe-M or Cr-M can be used in HEA with dominant Ti content; however, the alloy composition should include iron and chromium. It should be noted that the calculated values of Md_Fe_ and Md_Cr_ for the chemical ingredients shown in Table 2 have values larger than 1.1, which is incompatible with the values in the cited paper (Md_Fe_ = 0.92; Md_Cr_ = 0.86). 

The performed analysis and assumptions of the Bo–Md model suitability for the estimation of the multicomponent alloys phase structure, including high-entropy alloys, were confirmed by the X-ray phase analysis. However, due to the multicomponent nature of the investigated alloy and the related varying degree of radiation absorption, specific lamps with two radiation lengths were used to precisely identify the crystal structures. Regardless of the radiation used, for both wavelengths the obtained reflections were pre-adjusted to the complex body-centered cubic (CBCC) manganese lattice with the parameter a = 8.9121 Å (based on the PDF 01-089-2133 sheet) and the hexagonal phase of Laves C14-TiFe2 with the parameters a = 4.7929 Å and c = 7.8234 Å (based on PDF sheet 04-004-8431) (Figure 10).

Due to the different radiation absorption of the alloy elements used and in order to accurately identify the obtained reflections, the diffractograms were related to the common parameter, namely the interplanar distance d_hkl_ (Figure 11a). This allowed for the indication of plane reflections (Figure 11b) and the determination of the changes in the lattice parameters for the identified phases as a function of the annealing time (Figure 12). The parameters of the cubic lattice determined by the quenching method fluctuate around the manganese lattice parameter, differing by ±0.6 Å (Figure 12a). One can conclude that the resulting solid solution was formed on the basis of the manganese lattice. In the case of the hexagonal lattice, the values of the lattice parameters and the ratio of the identity periods c/a oscillating around the value 1.63 confirmed the formation of the multicomponent, intermetallic Laves C14 phase (Figure 12b,c). It should be highlighted that the changes in the lattice parameters correspond to the changes in the chemical composition. In both analyzed cases, it was observed that along with intense changes in the chemical composition observed after about 10 h of annealing, there is a nonmonotonic change in the lattice parameters of both identified phases. This allows us to distinguish the presence of three time intervals: an initial incubation period of up to 10 h of annealing, driven by the effect of slow diffusion, which is typical for high-entropy alloys; a period of significant fluctuations in both the chemical composition and lattice parameters; and finally, the stabilization of the content of individual alloy components and phase structure. 

## 4. Conclusions

The main concluding remarks of the present work are presented below. 

The molecular orbitals and equilibrium systems-based methodology led to the successful development of a Bo–Md diagram that further enables us to predict the multiphase structure of high-entropy alloys at the stage of their design;The sintering process enabled us to form an ultra-fine-grained, two-phase TiCoCrFeMn alloy consisting of the BCC solid solution and the Laves C14 phase;The slow diffusion effect, typical for high-entropy alloys, results in a long incubation period of up to 10 h at 1000 °C, preceding the subsequent changes in the chemical composition;The evolution of the chemical composition occurred during homogenization at 1000 °C for up to 1000 h, observed using the EDS method, and its impact on changes in the parameters of the crystal lattices of the identified phases are reflected in the Bo–Md diagram. One could confirm an appropriate sensitivity of the molecular orbitals model in the area of a multicomponent, multiphase alloys structural analysis.It was found that the description of the precipitation phenomena using the theory of molecular orbitals must consider the simultaneous changes in the parameters characterizing the forces of covalent bonding between the alloying elements used (Bo parameter) and the average energy level of the d-orbital (parameter Md), which must be determined for the dominant alloy element.

## Figures and Tables

**Figure 1 materials-14-05285-f001:**
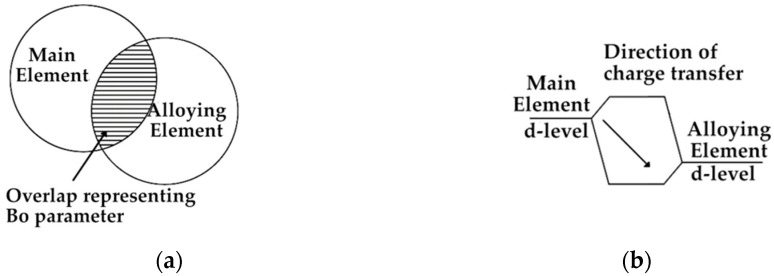
Graphical representation of Bo (**a**) and Md (**b**) parameters.

**Figure 2 materials-14-05285-f002:**
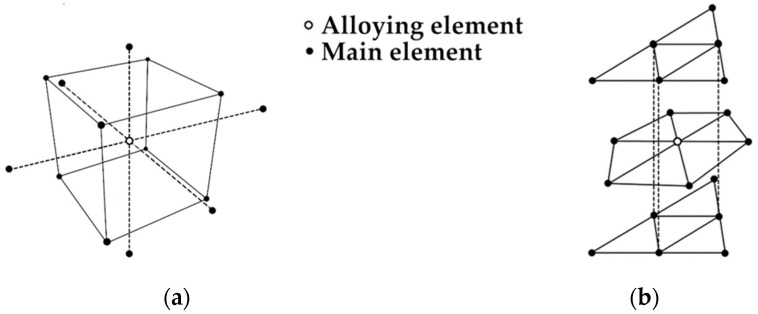
Placement of the main and alloying element atoms in body-centered cubic lattice and (**a**) hexagonal close-packed lattice (**b**).

**Figure 3 materials-14-05285-f003:**
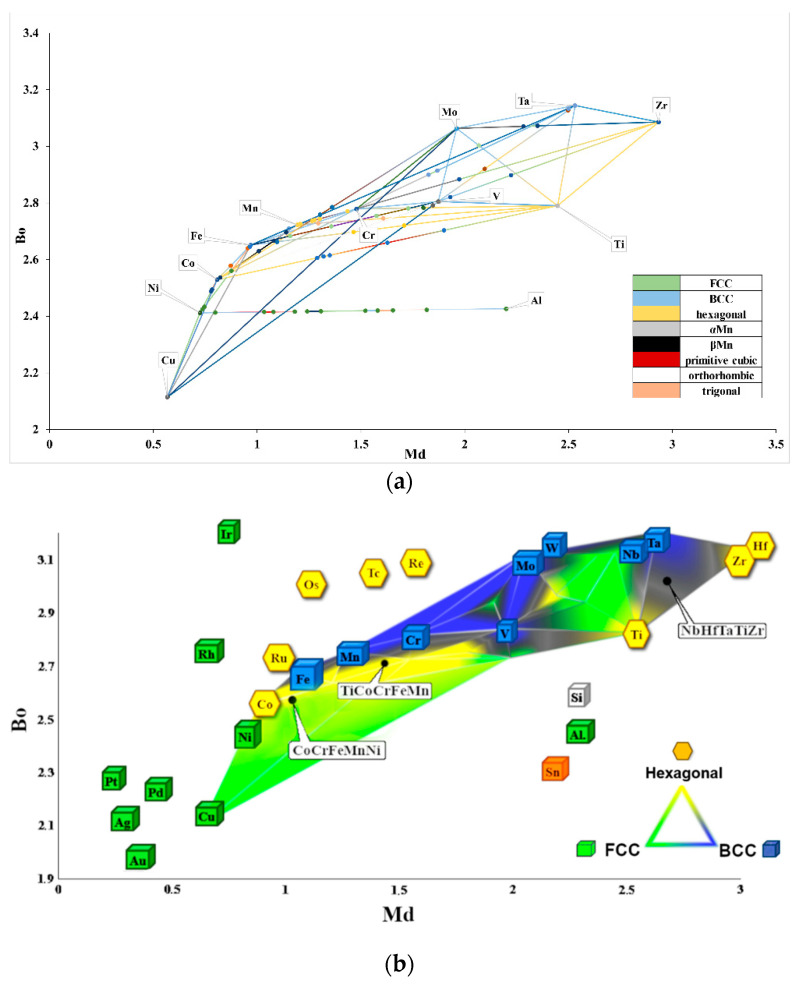
The Bo–Md diagram developed using the molecular orbitals method for elements based on the FCC, BCC and hexagonal lattices together with the position of the designed TiCoCrFeMn alloy in the BCC + hexagonal two-phase area, the Cantor CoCrFeMnNi alloy with the FCC lattice and the NbHfTaTiZr alloy with the BCC lattice [3]; construction lines compatible with double equilibrium systems (**a**) and the area of dominant phases with BCC, FCC and hexagonal structure (**b**).

**Figure 4 materials-14-05285-f004:**
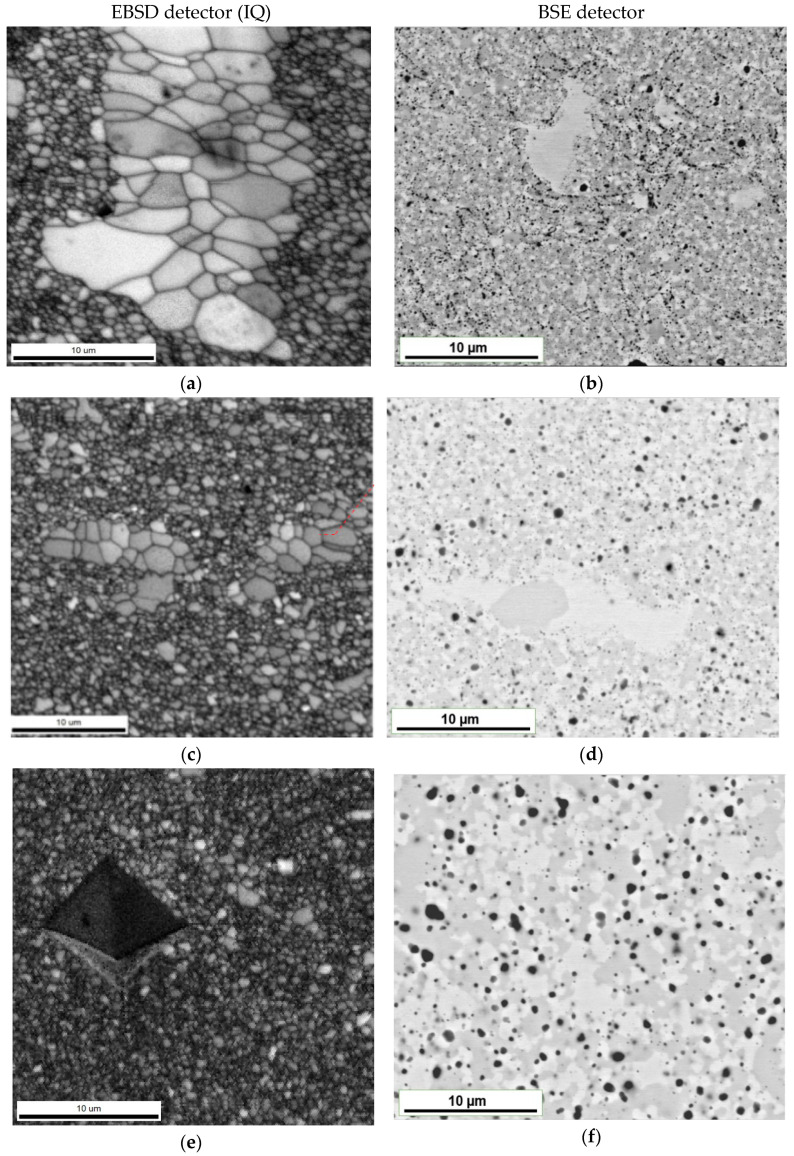
Two-phase microstructure of TiCoCrFeMn alloy with large bright areas disappearing as a function of homogenization time after annealing during 1 h (**a**,**b**), 100 h (**c**,**d**) and 1000 h (**e**,**f**).

**Figure 5 materials-14-05285-f005:**
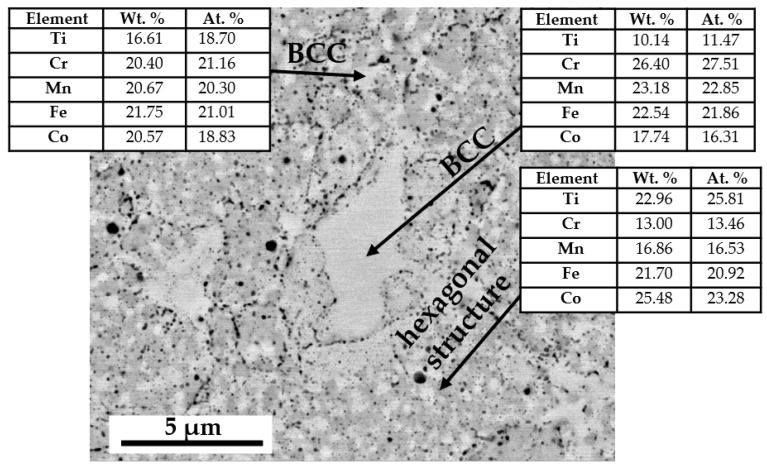
Microanalysis of the TiCoCrFeMn alloy chemical composition after initial sintering in bright areas (BCC lattice-based elements) and grey areas with increased content of elements with a hexagonal lattice.

**Figure 6 materials-14-05285-f006:**
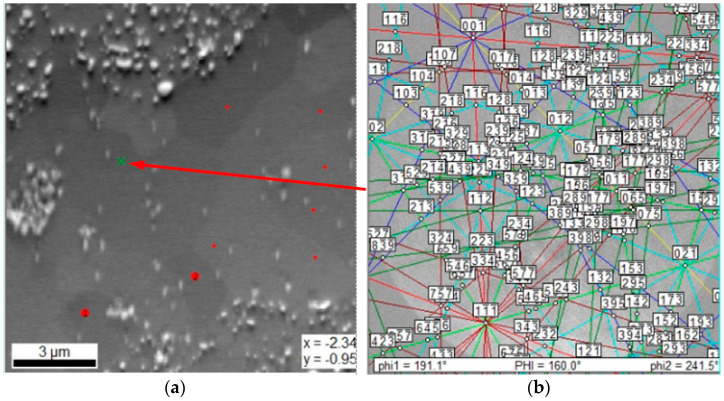
An example of a large light phase area (lb) subjected to EBSD microdiffraction (**a**) in the form of the Kikuchi line for the BCC crystal lattice with the lattice parameter oscillating around the value characteristic for the manganese lattice (**b**).

**Figure 7 materials-14-05285-f007:**
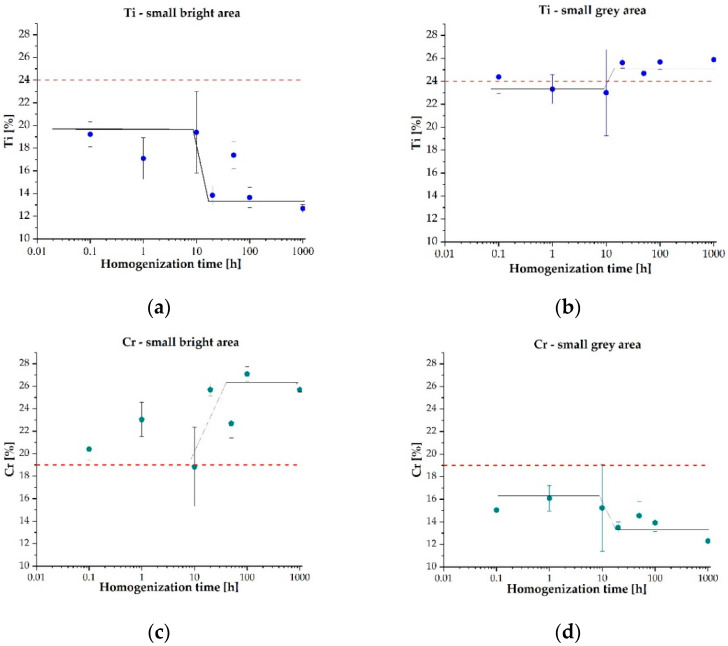
Evolution of the chemical composition of titanium (**a**,**b**) and chromium (**c**,**d**) in the areas of the two-phase, ultra-fine-grained matrix of the TiCoCrFeMn sinter as a function of homogenization time at 1000 °C.

**Figure 8 materials-14-05285-f008:**
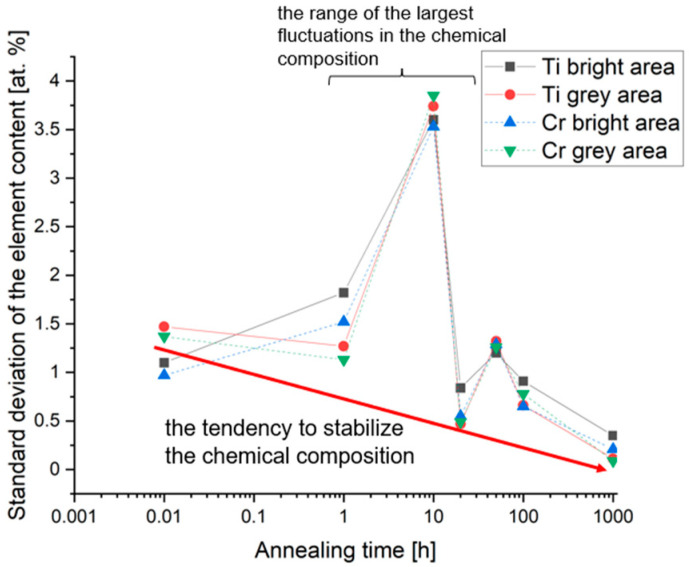
Nonmonotonic change in the stabilization tendency of the chromium and titanium chemical composition in the areas of the two-phase, ultra-fine-grained matrix of the TiCoCrFeMn sinter, observed in the area of 10 h of homogenization at 1000 °C.

**Figure 9 materials-14-05285-f009:**
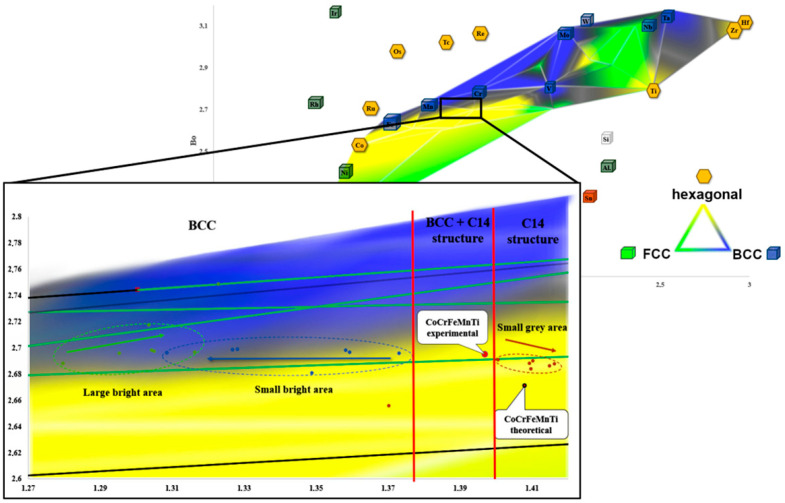
The effect of homogenization time on the chemical composition evolution of the two-phase sinter structure TiCoCrFeMn presented using the Bo–Md diagram.

**Figure 10 materials-14-05285-f010:**
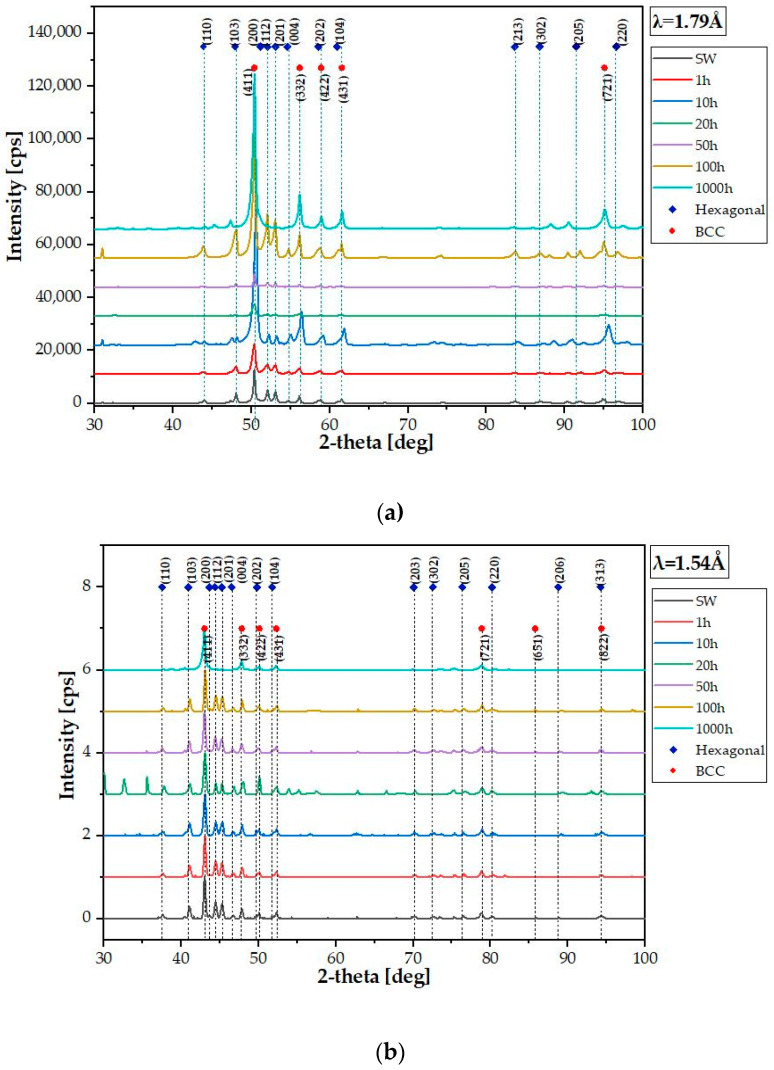
TiCoCrFeMn alloy diffractograms after different annealing times for the two used X-ray wavelengths, Co lamp (**a**), Cu lamp (**b**).

**Figure 11 materials-14-05285-f011:**
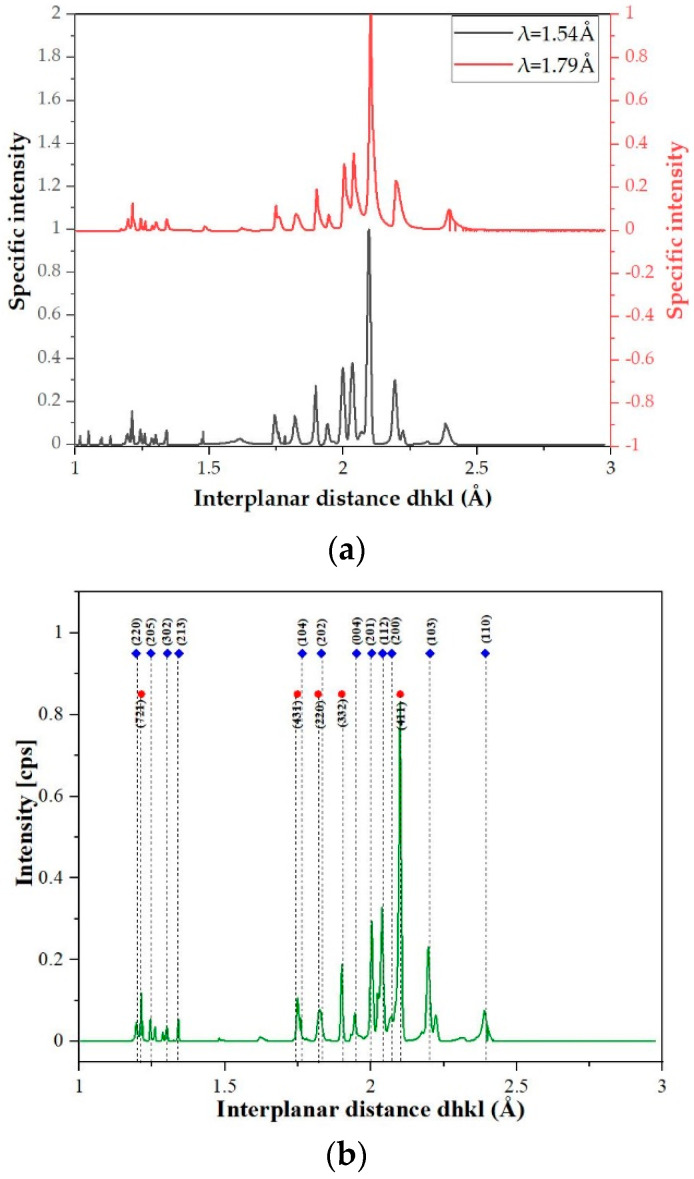
Exemplary diffractograms of TiCoCrFeMn alloy annealed during 100 h presented as a function of the interplanar distance d_hkl_ (**a**) and the result of indicating the repetitive reflections (**b**).

**Figure 12 materials-14-05285-f012:**
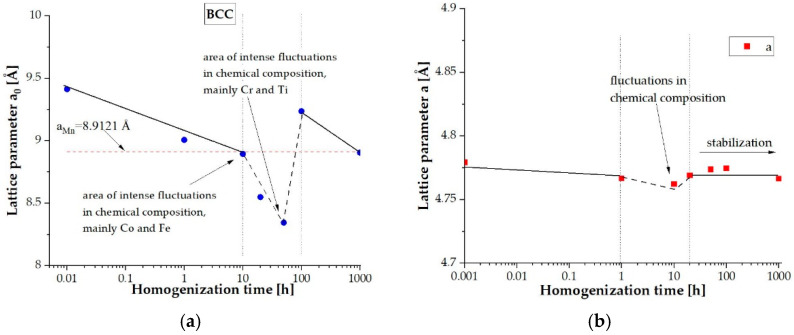
Changes in the parameters of the BCC (**a**) solid solution lattice and the Laves C14 (**b**–**d**) phase as a function of the homogenizing annealing time.

**Table 1 materials-14-05285-t001:** Chemical composition of the high-entropy alloy in question given in atomic percent.

Element	Ti	Co	Cr	Fe	Mn
At. %	23.6	19.1	19.1	19.1	19.1

**Table 2 materials-14-05285-t002:** The contribution of individual elements (at. %) with the standard deviation for three observed phase areas: large bright area (lb), small bright area (sb) and small grey area (sg) of TiCoCrFeMn sintered and annealed at 1000 °C alloy.

		Preliminary Sintering	1 h	10 h	20 h	50 h	100 h	1000 h
Ti	lb	13.34 ± 0.92	11.81 ± 0.21	11.31 ± 0.20	11.96 ± 0.12	11.66 ± 0.10	11.85 ± 0.13	11.49 ± 0.12
	sb	19.23 ± 1.10	17.09 ± 1.82	19.40 ± 3.60	13.85 ± 0.84	17.39 ± 1.20	13.65 ± 0.91	12.68 ± 0.35
	sg	24.38 ± 1.47	23.31 ± 1.27	23.00 ± 3.74	25.60 ± 0.47	24.68 ± 1.32	25.68 ± 0.66	25.87 ± 0.11
Co	lb	16.35 ± 0.59	16.74 ± 2.92	21.49 ± 0.22	18.50 ± 4.01	16.50 ± 0.10	16.89 ± 0.06	17.27 ± 0.05
	sb	18.43 ± 0.52	17.25 ± 0.74	25.13 ± 2.11	16.43 ± 0.25	17.93 ± 0.68	16.79 ± 0.24	17.16 ± 0.13
	sg	21.68 ± 0.82	21.34 ± 0.56	26.34 ± 3.83	22.84 ± 0.36	22.76 ± 0.62	23.57 ± 0.43	24.42 ± 0.07
Cr	lb	25.20 ± 1.58	25.20 ± 2.36	25.83 ± 0.58	26.62 ± 0.17	27.06 ± 0.25	27.48 ± 0.13	26.29 ± 0.18
	sb	20.41 ± 0.97	23.04 ± 1.52	18.84 ± 3.53	25.69 ± 0.55	22.69 ± 1.29	27.09 ± 0.65	25.68 ± 0.21
	sg	15.04 ± 1.37	16.09 ± 1.13	15.24 ± 3.85	13.50 ± 0.49	14.55 ± 1.26	13.92 ± 0.78	12.31 ± 0.09
Fe	lb	23.46 ± 2.19	24.56 ± 2.87	20.53 ± 0.37	21.75 ± 0.14	21.67 ± 0.27	22.31 ± 0.07	22.51 ± 0.10
	sb	21.60 ± 0.35	21.56 ± 0.19	19.18 ± 0.77	21.70 ± 0.15	21.06 ± 0.11	22.35 ± 0.14	22.62 ± 0.10
	sg	21.39 ± 0.50	21.67 ± 0.21	18.52 ± 0.50	21.56 ± 0.18	20.99 ± 0.17	21.90 ± 0.14	21.73 ± 0.08
Mn	lb	21.62 ± 0.95	22.52 ± 0.62	20.83 ± 0.38	23.17 ± 0.12	23.11 ± 0.16	21.47 ± 0.10	22.45 ± 0.08
	sb	20.33 ± 0.65	21.05 ± 0.88	17.34 ± 1.44	22.33 ± 0.31	20.91 ± 0.56	20.12 ± 0.33	21.86 ± 0.13
	sg	17.51 ± 0.50	17.60 ± 0.58	15.92 ± 1.47	16.50 ± 0.21	17.02 ± 0.66	14.93 ± 0.29	15.67 ± 0.08

## Data Availability

The data are available in a publicly accessible repository.

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
