# Peer review of "TiCoCrFeMn (BCC + C14) High-Entropy Alloy Multiphase Structure Analysis Based on the Theory of Molecular Orbitals"

_materials, 2021, doi:10.3390/ma14185285_

Round 1
Reviewer 1 Report
Authors have addressed majority of suggested corrections and modifications. There is one additional clarification which has to be made before the manuscript can be accepted for publication.
1. Throughout the text, the authors use "EBSD X-ray microdiffraction" as a designation for experimental method. However, I am not familiar with the experimental technique, where the electron backscatter diffraction detector or EBSD as a method would be working with the X-ray signal. EBSD detector works with the electron diffraction and what authors show in Figure 11 are indexed Kikuchi bands in electron backscatter diffraction pattern.
I assume that by using term "EBSD X-ray microdiffraction" the authors reference to possible "integrated EBSD/EDS mapping" where the area is scanned in SEM and signal is collected by both EBSD and EDS detectors for every pixel.
The terminology has to be corrected/clarified throughout the text, or the authors has to provide literature reference and explanation for the method called "EBSD X-ray microdiffraction".
Author Response
Detailed Response to Reviewer Comments
Ms. Ref. No.: materials-1359224
Title: Novel design concept of high - entropy alloys with multiphases structures
Materials
Dear Sir or Madame,
I would like to thank you very much for your letter and the reviewer’s comments on our manuscript (No.: materials-1359224). We appreciate your very valuable comments, that gave us a chance for revising the manuscript.
We have addressed all of the comments and revised the manuscript accordingly. Detailed responses to the comments are described in the “Response to Reviewers” point by point with line numbers given in each answer.
We now resubmit the manuscript for your further consideration for publication in your journal. We sincerely hope this revised manuscript will be finally acceptable for publication. If you have any questions about this manuscript, please do not hesitate to contact me.
Best regards
Dominika Górniewicz
On behalf of all co-authors
Military University of Technology
E-mail: dominika.gorniewicz@wat.edu.pl
Reviewer’s Comments:
Reviewer #1:
Review 1:
Authors have addressed majority of suggested corrections and modifications. There is one additional clarification which has to be made before the manuscript can be accepted for publication.
- Throughout the text, the authors use "EBSD X-ray microdiffraction" as a designation for experimental method. However, I am not familiar with the experimental technique, where the electron backscatter diffraction detector or EBSD as a method would be working with the X-ray signal. EBSD detector works with the electron diffraction and what authors show in Figure 11 are indexed Kikuchi bands in electron backscatter diffraction pattern.
I assume that by using term "EBSD X-ray microdiffraction" the authors reference to possible "integrated EBSD/EDS mapping" where the area is scanned in SEM and signal is collected by both EBSD and EDS detectors for every pixel.
The terminology has to be corrected/clarified throughout the text, or the authors has to provide literature reference and explanation for the method called "EBSD X-ray microdiffraction".
Response: We would like to thank the reviewer for the comment. EBSD observations were made using backscattered electrons. A stationary primary beam hits the sample, and the diffracting electrons create a diffraction pattern on a fluorescent screen that is covered with a phosphor. In this case we did not perform the "integrated EBSD/EDS mapping" as the reviewer suggest. The experimental method described in this article as "EBSD X-ray microdiffraction" is incorrect because this method does not use X-rays. Corrections were applied in an article in 171-172, 266, and 297 lines.

Reviewer 2 Report
The authors used a modified Bo-Md method to predict the multi-phase structure of a CoCrFeMnTi high-entropy alloy (HEA). This was combined with experimental measurements to confirm the multiple phases present in this alloy. While the experimental part looks rich and decent, the proposed modified Bo-Md method was poorly introduced. Besides this, there are other major and minor issues that prevent the current manuscript from being accepted. I will walk you through these issues step by step as follows.
Major issues:
1. The scientific background of the Bo-Md method needs to be articulated. For example, what are the physical meanings of changing the values of Bo and Md, and how do we imply from their values the crystal structures of a given metal or alloy? I know there are potential references cited here for these, but making these explicit is critical to the impact of the present work. This is a very common issue found in reviewing manuscripts based on my experience.
2. In the modified Bo-Md method, how did the authors determine the areas for different crystal structures as well as the boundaries of these areas? How are the Bo and Md values of alloys calculated? Only FCC, BCC, and HCP are available from this method (Fig. 2), but how do you know that the predicted structure corresponds to a solid solution or Laves phase?
3. Throughout the manuscript, the authors used HCP structure for C14 Laves phase. This is erroneous. C14 is not HCP but only hexagonal. Another example is C15 which is cubic, and one should never say C15 is FCC. Please check the crystallographic definition of these structures. That being said, such a misunderstanding misleads the readers because the authors seemed to use C14 and HCP interchangeably in both the text and figures. The authors need to clarify these in detail, e.g. how do we imply C14 from Fig. 2? What should we expect changing HCP to C14 in Figs. 8 and 9? etc.
Minor issues:
1. Line 209: Fig. 3a, c, d should be Fig. 3a, c, e.
2. Line 234: Figure 1 should be Figure 2.
3. Lines 235 and 247 and Figure 4 caption: what is HZ lattice?
4. Line 250: what is chromium fluctuation? Please clarify this.
5. Figure 11: please introduce more details of the Kikuchi line system plot, such that the readers can learn to read useful information from this plot.
Author Response
Detailed Response to Reviewer Comments
Ms. Ref. No.: materials- 1359224
Title: Novel design concept of high - entropy alloys with multiphases structures
Materials
Dear Sir or Madame,
I would like to thank you very much for your letter and the reviewer’s comments on our manuscript (No.: materials- 1359224). We appreciate your very valuable comments, that gave us a chance for revising the manuscript.
We have addressed all of the comments and revised the manuscript accordingly. Detailed responses to the comments are described in the “Response to Reviewers” point by point with line numbers given in each answer.
We now resubmit the manuscript for your further consideration for publication in your journal. We sincerely hope this revised manuscript will be finally acceptable for publication. If you have any questions about this manuscript, please do not hesitate to contact me.
Best regards
Dominika Górniewicz
On behalf of all co-authors
Military University of Technology
E-mail: dominika.gorniewicz@wat.edu.pl
Reviewer’s Comments:
Reviewer #2:
The authors used a modified Bo-Md method to predict the multi-phase structure of a CoCrFeMnTi high-entropy alloy (HEA). This was combined with experimental measurements to confirm the multiple phases present in this alloy. While the experimental part looks rich and decent, the proposed modified Bo-Md method was poorly introduced. Besides this, there are other major and minor issues that prevent the current manuscript from being accepted. I will walk you through these issues step by step as follows.
Major issues:
- The scientific background of the Bo-Md method needs to be articulated. For example, what are the physical meanings of changing the values of Bo and Md, and how do we imply from their values the crystal structures of a given metal or alloy? I know there are potential references cited here for these, but making these explicit is critical to the impact of the present work. This is a very common issue found in reviewing manuscripts based on my experience.
Response: We would like to thank the reviewer for the comment. In this paper, we do not deal with the physical foundations of the molecular orbitals theory. This issue was developed by the authors of the works cited by us. In this article, we use their scientific achievements to develop a tool for predicting the phase structure on the example of HEA alloys.
- In the modified Bo-Md method, how did the authors determine the areas for different crystal structures as well as the boundaries of these areas? How are the Bo and Md values of alloys calculated? Only FCC, BCC, and HCP are available from this method (Fig. 2), but how do you know that the predicted structure corresponds to a solid solution or Laves phase?
Response: We would like to thank the reviewer for the comment. Authors determine the areas for different crystal structures based on the equilibrium systems. As we write in lines 201-206: The positions of individual elements on the system axes were determined using the literature data [27]. Subsequently, the points corresponding to the individual elements were connected with lines. Each line connecting these two elements corresponded to a double equilibrium system, indicating the areas of individual phases occurrence and characterized by a specific crystal lattice. Furthermore, Figure 2 a) shows how to create a graph. Each binary equilibrium system is a line on which the phases occurring are marked. As shown in Figure 2 a), there were several phases and the graph would therefore be unreadable. Therefore, the most common (also in HEA alloys) phases were selected: FCC, BCC and HCP. Also corrections were applied in an article in 319-342 lines and detailed response is located below.
Changes in chemical composition observed by using microscopic methods could be also observed on Bo-Md diagram (Figure 7). The calculations of covalent bond av-erage strength (Bo parameter) and average d-orbital energy level (Md parameter) for actual content of elements (Table 1) revealed locations of characteristic points. Those points and their relative movement on Bo-Md diagram in function of annealing time shows, that “grey” phase is moving towards HPC region and “small bright phase” is going in direction of BCC regions. Moreover, it could be observed, that diffusion oc-curred in large bright regions leads to changes in chemical composition. Those changes results in subsequent formation of solid solution with BCC structure characterized by the same parameters as for ultra-fine grained matrix. Such behavior was associated to homogenization in regions of solid solution formed during sintering process. This homogenization was performed in temperature of 1000°C and time equal to 10h. Such a long time required for incubation could be a proof of sluggish diffusion, characteristic for HEA. Thus, by using methodology proposed in work [36] it was assumed, that in investigated alloy TiCoCrFeMn border value MdTi for parameters Ti-M [28], below which solid solution with BCC structure occurred, equals to 1.375. On the other hand, Laves Phase with C14 structure are forming above 1.40 values of MdTi. The analysis Bo-Md diagrams from [29-35] allows to state, that Ti-based BCC structure occurs with Md values much higher, even above 2.5, which is evidently dependent of Bo values. Sheikh et al. [36] suggests, that border Md parameter, that characterizes precipitations in Fe-M or Cr-M can be used in HEA with dominant Ti content however, the alloy composition should include iron and chromium. It should be noted, that calculated values of MdFe and MdCr for chemical ingredients shown in Table 2 have values larger than 1.1 which is incompatible with values in cited paper (MdFe=0,92; MdCr=0,86).
- Throughout the manuscript, the authors used HCP structure for C14 Laves phase. This is erroneous. C14 is not HCP but only hexagonal. Another example is C15 which is cubic, and one should never say C15 is FCC. Please check the crystallographic definition of these structures. That being said, such a misunderstanding misleads the readers because the authors seemed to use C14 and HCP interchangeably in both the text and figures. The authors need to clarify these in detail, e.g. how do we imply C14 from Fig. 2? What should we expect changing HCP to C14 in Figs. 8 and 9? etc.
Response: We would like to thank the reviewer for the comment. Based on our knowledge and literature review, it is known that the Laves phase family consists of the three most important members. This is described as the C14 hexagonal MgZn2-type, the C15 cubic MgCu2-type and the C36 hexagonal MgNi2-type structures. The use of HCP and C14 interchangeably in this manuscript is incorrect, but you can use: hexagonal C14 structure or C14 structure instead. The above data was written based on references:
Gschneidner K. A., Jr., Pecharsky V. K., Binary rare earth Laves phases –– an overview, Z. Kristallogr. 2006, 221, 375–381.
Stein F., Leineweber A., Laves phases: a review of their functional and structural applications and an improved fundamental understanding of stability and properties, J Mater Sci 2021, 56, 5321–5427.
Minor issues:
- Line 209: Fig. 3a, c, d should be Fig. 3a, c, e.
Response: We would like to thank the reviewer for the comment. Changes have been made according to this comment.
- Line 234: Figure 1 should be Figure 2.
Response: We would like to thank the reviewer for the comment. Changes have been made according to this comment.
- Lines 235 and 247 and Figure 4 caption: what is HZ lattice?
Response: We would like to thank the reviewer for the comment. HZ lattice is the Polish abbreviation for the HCP lattice and this error has already been corrected in the manuscript.
- Line 250: what is chromium fluctuation? Please clarify this.
Response: We would like to thank the reviewer for the comment. Chromium fluctuation are non-monotonic changes in the content of this element.
- Figure 11: please introduce more details of the Kikuchi line system plot, such that the readers can learn to read useful information from this plot.
Response: We would like to thank the reviewer for the comment. In the EBSD experimental method, the primary electron beam hits the sample at a slight angle. Then backscattered electrons are diffracted by the crystal lattice of the sample to form cones Kossel. The intersections of the Kossel cones with the phosphor-coated screen form a Kikuchi line pattern unique to a given crystallite orientation. Kikuchi lines reflect geometrical relationships between crystallographic planes. The unique arrangement of these lines for a given crystallographic orientation allows for obtaining orientation maps and phase analysis of sample components. Bearing in mind the physical foundations of this method, lattice parameters data obtained from X-Ray Diffraction (XRD) analysis were declared, obtaining Kikuchi lines for the indicated crystallographic lattices.

Reviewer 3 Report
The manuscript “Novel design concept of high - entropy alloys with multiphases structures” by Gorniewicz et al. presents a high-entropy alloy (HEA) based on the Bo-Md parameters map. The study presents a new HEA – TiCoCrFeMn, fabricated by mechanical alloying and sintering method. The structure of the HEA was analyzed by XRD and SEM.
The study is based on a well-established Bo-Md map for designing Ti-based HEAs that has been shown to exhibit transformation-induced plasticity (TRIP) effect. The manuscript needs to be significantly improved and should include additional experiments that go beyond simply examining the structure. The SEM images provided in the paper are of poor quality and are not suitable to be published in a scientific journal. The paper simply presents the results, without any substantive discussion or comparison to the literature. The manuscript should include structure analysis details; at the very least, I was expecting confirmation of different phases by the TEM selected area diffraction patterns. Characterization of mechanical properties of sintered powders, such as hardness testing, compressive or tensile testing, is also required.
The specific issues with this manuscript are highlighted below.
- The abstract claims: “Thus, at the design stage of HEA fabrication, the method of molecular orbitals was proposed.”. The concept of designing HEAs using Bo-Md parameters is not new. Lilensten et al. [“Design and tensile properties of a bcc Ti-rich high-entropy alloy with transformation-induced plasticity,” Materials Research Letters, 5 (2017) 110–116] extended the work of Abdel-Hady et al. [“Predicting the solid solubility limit in high-entropy alloys using the molecular orbital approach,” Journal of Applied Physics, 118 (2015) 194902] for Bo-Md map of conventional Ti-alloys to Ti-rich HEAs. They designed a new HEA (Ti35Zr27.5Hf27.5Nb5Ta5) and demonstrated that the alloy undergoes transformation-induced plasticity (TRIP). Recently, Jung et al. [“Investigation of phase-transformation path in TiZrHf(VNbTa)x refractory high-entropy alloys and its effect on mechanical property,” Journal of Alloys and Compounds, 886 (2021) 161187] also used the same concept for the TRIP effect in TiZrHf(VNbTa)x HEAs. It goes back even further, to Sheikh et al. [“Predicting the solid solubility limit in high-entropy alloys using the molecular orbital approach,” Journal of Applied Physics, 118 (2015) 194902], who proposed using Bo-Md for HEAs. It is unethical to make this claim while ignoring a body of work focusing on the same problem. The article lacks a proper discussion of the previous work done for designing HEAs using Bo-Md parameters. Previous papers on designing HEAs based on the Bo-Md map went beyond merely creating a new HEA. By conducting deformation experiments and characterizing the microstructure during and after deformation, they demonstrated improved mechanical properties.
- The title of the paper is too broad and ambiguous. The manuscript focuses on one specific composition of HEA, i.e., TiCoCrFeMn. This type of title is reserved for review articles rather than original research papers. The title should be revised to be more specific and should not include the word “novel”, as HEAs have been extensively studied over the last two decades. There are also hundreds of papers on multiphase HEAs.
- The abstract of the paper begins with, “High-entropy alloys (HEA) are a group of modern construction materials…”; to the best of the reviewer’s knowledge, HEAs have no bulk applications as a construction material. The second sentence of the abstract says, “The complexity of their chemical composition and the further interactions of cobalt, chromium, iron, manganese and titanium based elements significantly inhibit the prediction of phases number that may form during material processing.” First, the authors must provide context for this sentence because they have yet to mention their alloy Ti-Co-Cr-Fe-Mn.
- Because this is not a Review Article, it is unnecessary to provide a figure indicating the number of HEA publications over the years (Fig. 1). Many review papers have similar figures. This figure should be removed. If authors want to highlight the rapid increase in the publication of HEAs (with different phases), they can do so in a single sentence and provide any of the references listed below:
-
- J. Pickering, N.G. Jones, “High-entropy alloys: a critical assessment of their founding principles and future prospects,” International Materials Reviews, 61 (2016) 183–202.
- B. Miracle, O.N. Senkov, “A critical review of high entropy alloys and related concepts,” Acta Materialia, 122 (2017) 448–511.
- Wang, W. Guo, Y. Fu, “High-entropy alloys: emerging materials for advanced functional applications,” Journal of Materials Chemistry A, 9 (2021) 663–701.
- Page-3: “It was assumed, that this alloy will be characterized by the designed phase structure, a combination of a solid solution and a Laves phase, which will be further verified by the Bo-Md diagram.” It is difficult to understand what this sentence is trying to convey. Throughout the paper, incorrect scientific terms are used, e.g., such as “participation of Laves phases” rather than ‘precipitation’ of Laves phase. Similarly, the term “Co lamp” and “Cu lamp” are used for Co and Cu targets in XRD. Instead of referring to phases with their proper names, they are referred to as “little bright phase area”, “small bright area” and so on.
- What’s the reason behind very large grains clustered together in the center of Fig. 3a and very fine grains in other regions?
- The authors included a Kikuchi line plot just before the conclusion. The results should be presented in a specific order. It is not appropriate to include another result at the end, with no details, and without an in-depth discussion.
- Conclusion#2: “The sintering process allowed to form an ultra-fine-grained two-phase TiCoCrFeMn alloy consisted of the assumed BCC solid solution and the Laves C14 phase”. Why is it an “assumed” BCC solid solution?
- Conclusion#3: “The effect of slow diffusion, typical for alloys with high entropy, caused intensive diffusion processes,”. There are contradictory reports about slow diffusion in HEAs; however, if it is accepted, how can “slow diffusion” have “caused intensive diffusion processes”?
- Conclusion#4: “The changes in the chemical composition observed with the use of classical research methodologies and their impact on changes in the parameters of the crystal lattices of the identified phases are reflected in the Bo-Md diagram, indicating the appropriate sensitivity of the proposed model in the area of structural analyzes.” First of all, it is not a new model proposed by the authors. Secondly, the changes occurring in composition with thermomechanical treatment are not anything astonishing. Thirdly, a conclusion should be a specific point derived from the results and discussion and not some generic statement.
Author Response
Detailed Response to Reviewer Comments
Ms. Ref. No.: materials-1359224
Title: Novel design concept of high - entropy alloys with multiphases structures
Materials
Dear Sir or Madame,
I would like to thank you very much for your letter and the reviewer’s comments on our manuscript (No.: materials-1359224). We appreciate your very valuable comments, that gave us a chance for revising the manuscript.
We have addressed all of the comments and revised the manuscript accordingly. Detailed responses to the comments are described in the “Response to Reviewers” point by point with line numbers given in each answer.
We now resubmit the manuscript for your further consideration for publication in your journal. We sincerely hope this revised manuscript will be finally acceptable for publication. If you have any questions about this manuscript, please do not hesitate to contact me.
Best regards
Dominika Górniewicz
On behalf of all co-authors
Military University of Technology
E-mail: dominika.gorniewicz@wat.edu.pl
Reviewer’s Comments:
Reviewer #3:
The manuscript “Novel design concept of high - entropy alloys with multiphases structures” by Gorniewicz et al. presents a high-entropy alloy (HEA) based on the Bo-Md parameters map. The study presents a new HEA – TiCoCrFeMn, fabricated by mechanical alloying and sintering method. The structure of the HEA was analyzed by XRD and SEM.
The study is based on a well-established Bo-Md map for designing Ti-based HEAs that has been shown to exhibit transformation-induced plasticity (TRIP) effect. The manuscript needs to be significantly improved and should include additional experiments that go beyond simply examining the structure. The SEM images provided in the paper are of poor quality and are not suitable to be published in a scientific journal. The paper simply presents the results, without any substantive discussion or comparison to the literature. The manuscript should include structure analysis details; at the very least, I was expecting confirmation of different phases by the TEM selected area diffraction patterns. Characterization of mechanical properties of sintered powders, such as hardness testing, compressive or tensile testing, is also required.
The specific issues with this manuscript are highlighted below.
Response: We would like to thank the reviewer for the comments and detailed review of our article. We would like to emphasize that this work concerns only and exclusively the use of the Bo-Md model in the field of analysis of phase transformations of multi-phase titanium based HEA alloys. The reviewer's comments turned our attention to the shortcomings in the article, which is why the revised work highlights the analysis of separation phenomena described by the Bo-Md diagram. We agree that the full characteristics of the material include the mutual dependence of the chemical composition, phase structure and the obtained properties. Nevertheless, the authors plan to present this area of issues connected with the stage of preparation of sintering powders in the next publication on this alloy.
- The abstract claims: “Thus, at the design stage of HEA fabrication, the method of molecular orbitals was proposed.”. The concept of designing HEAs using Bo-Md parameters is not new. Lilensten et al. [“Design and tensile properties of a bcc Ti-rich high-entropy alloy with transformation-induced plasticity,” Materials Research Letters, 5 (2017) 110–116] extended the work of Abdel-Hady et al. [“Predicting the solid solubility limit in high-entropy alloys using the molecular orbital approach,” Journal of Applied Physics, 118 (2015) 194902] for Bo-Md map of conventional Ti-alloys to Ti-rich HEAs. They designed a new HEA (Ti35Zr27.5Hf27.5Nb5Ta5) and demonstrated that the alloy undergoes transformation-induced plasticity (TRIP). Recently, Jung et al. [“Investigation of phase-transformation path in TiZrHf(VNbTa)x refractory high-entropy alloys and its effect on mechanical property,” Journal of Alloys and Compounds, 886 (2021) 161187] also used the same concept for the TRIP effect in TiZrHf(VNbTa)x HEAs. It goes back even further, to Sheikh et al. [“Predicting the solid solubility limit in high-entropy alloys using the molecular orbital approach,” Journal of Applied Physics, 118 (2015) 194902], who proposed using Bo-Md for HEAs. It is unethical to make this claim while ignoring a body of work focusing on the same problem. The article lacks a proper discussion of the previous work done for designing HEAs using Bo-Md parameters. Previous papers on designing HEAs based on the Bo-Md map went beyond merely creating a new HEA. By conducting deformation experiments and characterizing the microstructure during and after deformation, they demonstrated improved mechanical properties.
Response: We would like to thank the reviewer for the comment. We would like to thank you for drawing attention to the necessity to present this problem in a broader way, including the relevant literature references. It was not the intention of the authors to ascribe to themselves the authorship of the Bo-Md model and to use it for structural analyzes of materials. Corrections were applied in an article in 88-103 lines and detailed response is located below.
This method, patented in 1987 by Yukawa et al. [28], was described in [26] on the example of δ ferrite formation in ferritic steels, which also consider hydrogen absorption and desorption in metal alloys. The correctness of this method for the analysis of the martensitic transformation in titanium alloys was also confirmed in numerous studies analyzing phase transformations in conventional titanium alloys [29-32]. This concept was also used for the analysis of phase transformation induced plasticity (TRIP) occurring in high entropy alloys with dominant titanium content [33-35]. It should be noted, however, that these works concern the changes occurred in the con-ventional alloys with a solid solution structure without precipitation of other phases, such as the Laves phase. The problem of solubility limit analysis of the alloying elements in high entropic solid solutions was analyzed in [36], where the limit value of the Md parameter was determined for BCC solutions based on iron and chromium (MdFe = 0.92 and MdCr = 0.86, respectively). The authors of this study also suggest, that these values could also be used in high-entropy alloys with a dominant content of tita-nium, and additional iron and / or chromium in the alloy composition.2. The title of the paper is too broad and ambiguous. The manuscript focuses on one specific composition of HEA, i.e., TiCoCrFeMn. This type of title is reserved for review articles rather than original research papers. The title should be revised to be more specific and should not include the word “novel”, as HEAs have been extensively studied over the last two decades. There are also hundreds of papers on multiphase HEAs.
Response: We would like to thank the reviewer for the comment. We fully agree with the reviewer's comment. The title of the article was changed to be more appropriate to the content of the work. Corrections were applied in an article detailed response is located below.
TiCoCrFeMn (BCC + C14) High-Entropy Alloy multiphase structure analysis based on the theory of molecular orbitals
- The abstract of the paper begins with, “High-entropy alloys (HEA) are a group of modern construction materials…”; to the best of the reviewer’s knowledge, HEAs have no bulk applications as a construction material. The second sentence of the abstract says, “The complexity of their chemical composition and the further interactions of cobalt, chromium, iron, manganese and titanium based elements significantly inhibit the prediction of phases number that may form during material processing.” First, the authors must provide context for this sentence because they have yet to mention their alloy Ti-Co-Cr-Fe-Mn.
Response: We would like to thank the reviewer for the comment. Corrections were applied in an article in 14-17 lines and detailed response is located below.
High-entropy alloys (HEA) are a group of modern, perspective materials, that have been intensively developed in recent years due to their superior properties and potential applications in many fields. The complexity of their chemical composition and the further interactions of main elements significantly inhibit the prediction of phases, that may form during material processing.
- Because this is not a Review Article, it is unnecessary to provide a figure indicating the number of HEA publications over the years (Fig. 1). Many review papers have similar figures. This figure should be removed. If authors want to highlight the rapid increase in the publication of HEAs (with different phases), they can do so in a single sentence and provide any of the references listed below:
- J. Pickering, N.G. Jones, “High-entropy alloys: a critical assessment of their founding principles and future prospects,” International Materials Reviews, 61 (2016) 183–202.
- B. Miracle, O.N. Senkov, “A critical review of high entropy alloys and related concepts,” Acta Materialia, 122 (2017) 448–511.
- Wang, W. Guo, Y. Fu, “High-entropy alloys: emerging materials for advanced functional applications,” Journal of Materials Chemistry A, 9 (2021) 663–701.
Response: We would like to thank the reviewer for the comment. We agree that this is not a review article, nevertheless the graph (Fig. 1) shows a huge disproportion between single-phase and multiphase alloys, proving the validity of the subject matter proposed in this article.
- Page-3: “It was assumed, that this alloy will be characterized by the designed phase structure, a combination of a solid solution and a Laves phase, which will be further verified by the Bo-Md diagram.” It is difficult to understand what this sentence is trying to convey. Throughout the paper, incorrect scientific terms are used, e.g., such as “participation of Laves phases” rather than ‘precipitation’ of Laves phase. Similarly, the term “Co lamp” and “Cu lamp” are used for Co and Cu targets in XRD. Instead of referring to phases with their proper names, they are referred to as “little bright phase area”, “small bright area” and so on.
Response: We would like to thank the reviewer for the comment. Regarding the nomenclature of the phases: "little bright phase area", "small bright area", the corresponding explanation is in the text. Corrections were made at the request of previous reviewers. Other corrections were applied in an article in 138-141, 42-44, 173-178 lines and detailed response is located below.
It was assumed, that designed alloy will have a two-phase structure consisting of a BCC solid solution and a Laves phase, and such structure will be verified by classical material characterization methods (SEM, XRD) and the proposed Bo-Md diagram.
Materials with the participation of the Laves phase, as the second component of the phase structure, are of particular interest among multi-phase HEA.
The phase fractions in the material were determined by using X-ray Diffraction (XRD) analysis on Rigaku X-Ray Diffractometer Ultima IV (Rigaku, Tokyo, Japan) with a Co lamp (λ = 1.79 Å) and PDF-4 database. The XRD measurements were repeated on Sei-fert XRD 3003 TT diffractometer (Seifert, Mannheinm, Germany) with a Cu lamp (λ = 1.54 Å) to identify plane reflections due to the different absorption of radiation of the elements.
- What’s the reason behind very large grains clustered together in the center of Fig. 3a and very fine grains in other regions?
Response: We would like to thank the reviewer for the comment. In terms of grain size analysis and its influence on mechanical properties (HV, KIc), the authors plan another work. Nevertheless, we want to clarify that the locally occurring "large grains" in sizes up to 5 µm are caused by the coagulation of chromium particles during the mechanical alloying stage. This effect will be presented in the planned next article. We will not explain this phenomenon in this paper because it only concerns phase phenomena in conjunction with the Bo-Md diagram.
- The authors included a Kikuchi line plot just before the conclusion. The results should be presented in a specific order. It is not appropriate to include another result at the end, with no details, and without an in-depth discussion.
Response: We would like to thank the reviewer for the comment. The placement of Kikuchi line plot just before the conclusion was caused to a comment from one of the reviewers. Initially, the analysis confirming the BCC structure in large bright areas was placed after Figure 4, which proved the correctness of the identification of the observed phase structure. Therefore, according to your suggestion, this drawing has been moved to a previous, more appropriate place. Corrections were applied in an article in 264-270 lines and detailed response is located below.
Nevertheless, in order to identify the crystallographic structure of the observed areas with different phase contrast, EBSD microdiffraction studies were performed in the areas of "large bright particles" on the basis of the crystal lattice parameters values ob-tained from the XRD tests (Figure 4). The obtained Kikuchi line system clearly indicate, that the bright phase has the structure of a BCC with a lattice constantly oscillating around the value characteristic for the manganese lattice
- Conclusion#2: “The sintering process allowed to form an ultra-fine-grained two-phase TiCoCrFeMn alloy consisted of the assumed BCC solid solution and the Laves C14 phase”. Why is it an “assumed” BCC solid solution?
Response: We would like to thank the reviewer for the comment. Corrections were applied in an article in 398-399 lines and detailed response is located below.
The sintering process enabled to form an ultra-fine-grained two-phase Ti-CoCrFeMn alloy consisted of the BCC solid solution and the Laves C14 phase.
- Conclusion#3: “The effect of slow diffusion, typical for alloys with high entropy, caused intensive diffusion processes,”. There are contradictory reports about slow diffusion in HEAs; however, if it is accepted, how can “slow diffusion” have “caused intensive diffusion processes”?
Response: We would like to thank the reviewer for the comment. Corrections were applied in an article in 400-402 lines and detailed response is located below.
The slow diffusion effect, typical for high entropy alloys, results in a long incuba-tion period of up to 10 hours at 1000 °C, preceding the subsequent changes in the chemical composition.
- Conclusion#4: “The changes in the chemical composition observed with the use of classical research methodologies and their impact on changes in the parameters of the crystal lattices of the identified phases are reflected in the Bo-Md diagram, indicating the appropriate sensitivity of the proposed model in the area of structural analyzes.” First of all, it is not a new model proposed by the authors. Secondly, the changes occurring in composition with thermomechanical treatment are not anything astonishing. Thirdly, a conclusion should be a specific point derived from the results and discussion and not some generic statement.
Response: We would like to thank the reviewer for the comment. At no point in the paper the authors claim that this is the model proposed by the authors. In addition, in this study, no thermomechanical treatment was used, but only the classic heat treatment consisting in long-term homogenization in 1000°C /100h aimed at obtaining equilibrium structures. This conclusion is based on analyzes of chemical composition changes, XRD observations and is also visualized in the Bo-Md plotted graph. Corrections were applied in an article in 403-407 lines and detailed response is located below.
Evolution of the chemical composition occurred during homogenization at 1000°C for up to 1000 hours, observed using the EDS method, and its impact on changes in the parameters of the crystal lattices of the identified phases, are reflected in the Bo-Md diagram. One could confirm an appropriate sensitivity of the molecular orbitals model in the area of multicomponent multiphase alloys structural analysis.

Round 2
Reviewer 2 Report
I appreciate the authors' responses to my comments. At the end of the day, my major concern with how the C14 phase formation was predicted from the modified Bo-Md plot has not been addressed.
First of all, as I said, HCP and C14 are different structures, and at most one can only say hexagonal C14. On the one hand, there are still multiple places in the main text saying e.g. "HCP lattice of the Laves phase". On the other hand, there are a number of experimental plots presenting the Laves phase as HCP. All of these places are scientifically wrong.
Second, even with the revised Bo-Md plots, it is still impossible to imply the formation of C14 Laves phase. The plot says that the HCP phase will form, but in reality it's a C14 Laves phase. Again, it should be emphasized that C14 does not have a HCP lattice, but only a hexagonal lattice. Such a discontinuity between the prediction and the experiment makes this modified approach broken.
While I'm fine with the authors' argument that the scientific foundations of Bo-Md are not center of this manuscript, the fact is that when the general audience who have had no previous experience with Bo-Md come to read your paper, they may get heavily hung up and disoriented from reading this paper. This is why slight but not too detailed introduction of this method is recommended.
The Kikuchi line plot is never referred to in the main text (Figure 11), and this figure is placed weirdly in-between Table 2 and Figure 5. What I meant was to introduce this plot more in the main text, not to me. At this moment, only one sentence throughout the manuscript is about this figure. The authors should clarify what the arrow, the crosses, and all kinds of colored lines and indices mean in this plot.
Author Response
Detailed Response to Reviewer Comments
Ms. Ref. No.: materials- 1359224
Title: TiCoCrFeMn (BCC + C14) High-Entropy Alloy multiphase structure analysis based on the theory of molecular orbitals
Materials
Dear Sir or Madame,
I would like to thank you very much for your letter and the reviewer’s comments on our manuscript (No.: materials- 1359224). We appreciate your very valuable comments, that gave us a chance for revising the manuscript.
We have addressed all of the comments and revised the manuscript accordingly. Detailed responses to the comments are described in the “Response to Reviewers” point by point with line numbers given in each answer.
We now resubmit the manuscript for your further consideration for publication in your journal. We sincerely hope this revised manuscript will be finally acceptable for publication. If you have any questions about this manuscript, please do not hesitate to contact me.
Best regards
Dominika Górniewicz
On behalf of all co-authors
Military University of Technology
E-mail: dominika.gorniewicz@wat.edu.pl
Reviewer’s Comments:
Reviewer #2:
- I appreciate the authors' responses to my comments. At the end of the day, my major concern with how the C14 phase formation was predicted from the modified Bo-Md plot has not been addressed.
First of all, as I said, HCP and C14 are different structures, and at most one can only say hexagonal C14. On the one hand, there are still multiple places in the main text saying e.g. "HCP lattice of the Laves phase". On the other hand, there are a number of experimental plots presenting the Laves phase as HCP. All of these places are scientifically wrong.
Second, even with the revised Bo-Md plots, it is still impossible to imply the formation of C14 Laves phase. The plot says that the HCP phase will form, but in reality it's a C14 Laves phase. Again, it should be emphasized that C14 does not have a HCP lattice, but only a hexagonal lattice. Such a discontinuity between the prediction and the experiment makes this modified approach broken.
Response: The reviewer's comment is correct. The presence of HCP lattice markings results from the oversight. According to this comment, appropriate corrections were made in the text and in the figures, where the HCP lattice was changed to hexagonal lattice.
- While I'm fine with the authors' argument that the scientific foundations of Bo-Md are not center of this manuscript, the fact is that when the general audience who have had no previous experience with Bo-Md come to read your paper, they may get heavily hung up and disoriented from reading this paper. This is why slight but not too detailed introduction of this method is recommended.
Response: Figures 1a, b and 2a, b have been added and corrections were applied in an article in 192-214 lines and detailed response is located below.
Molecular orbital method, called Bo-Md consists of calculations of two parameters for main element in alloy. Bo parameter (visualization on Figure 1a) is short from bond order and describes strength of bonding between main element and alloying element. Bond order value differs for different crystal structures as shown in Figure 2. Alloying element has different number of neighbours in different distance in cubic, body centred lattice (Figure 2a) and different in hexagonal close packaged lattice (Figure 2b). Md parameter (Figure 1b) is short from metal d-orbital energy and connects electronegativity and atomic radius of alloying elements together with main element in alloy. Energy level in hybridized bond between main and alloying element controls charge transfer direction, so it is related to electronegativity. Main higher element of d-level energy (Figure 1b) means that charge transfer is from main to alloying element. Md parameter is affected by the atomic radius thus the radius of d orbital increased and subsequently, its energy level is higher [25]. The calculations of both parameters for complex alloy were presented below:
Equation are available in the file attached. |
(1) |
Equation are available in the file attached. |
(2) |
Where, Xi is the atomic fraction of component i in the alloy, and (Md)i and (Bo)i are values for i component.
- The Kikuchi line plot is never referred to in the main text (Figure 11), and this figure is placed weirdly in-between Table 2 and Figure 5. What I meant was to introduce this plot more in the main text, not to me. At this moment, only one sentence throughout the manuscript is about this figure. The authors should clarify what the arrow, the crosses, and all kinds of colored lines and indices mean in this plot.
Response: Corrections were applied in an article in 272-275 lines and detailed response is located below.
The cross in Figure 6a indicates the area of the analysis in which the electron diffraction recorded (Figure 6b). The Kikuchi lines shown in Figure 6b were indexed according to the declared lattice parameters. Additionally, red dots indicate the sites subjected to the analysis with the same parameters as these point with a cross.

Reviewer 3 Report
I am glad to see that the authors took the comments seriously and attempted to make corrections in the manuscript. I see that it has improved in many areas, but I feel they should address the following issues before the paper can be accepted.
The second author of this paper is from GeniCore (https://genicore.eu/), a company that sells Spark Plasma Sintering equipment, materials (including high-entropy alloys), and also provides related services. The authors stated that they did not have any conflicts of interest. This affiliation appears to be a conflict of interest which should have been disclosed in the “Conflict of Interest” statement. Authors should explicitly declare this in their conflict of interest statement.
In response to my comment on Fig. 1, it appears that the authors did not take it seriously. This figure adds no value to their work. They should recognize that their approach is flawed. You obtained information about the number of publications for single-phase and multiphase HEAs from the title of papers, and as a result, you may have underestimated the number of publications with multiphase, for example, the first papers on HEAs by both Cantor and Yeh included alloys with multiple phases, which cannot be counted simply by looking at the titles of their papers. The reason that titles on HEAs appear to be more focused on single-phase HEAs is because single-phase HEAs are difficult to obtain and multiphase alloys are more likely to form (again, see Cantor's first paper). However, you will not be able to obtain that information by searching the database. The authors must remove this figure and follow my advice from the previous report.
Author Response
Detailed Response to Reviewer Comments
Ms. Ref. No.: materials- 1359224
Title: TiCoCrFeMn (BCC + C14) High-Entropy Alloy multiphase structure analysis based on the theory of molecular orbitals
Materials
Dear Sir or Madame,
I would like to thank you very much for your letter and the reviewer’s comments on our manuscript (No.: materials- 1359224). We appreciate your very valuable comments, that gave us a chance for revising the manuscript.
We have addressed all of the comments and revised the manuscript accordingly. Detailed responses to the comments are described in the “Response to Reviewers” point by point with line numbers given in each answer.
We now resubmit the manuscript for your further consideration for publication in your journal. We sincerely hope this revised manuscript will be finally acceptable for publication. If you have any questions about this manuscript, please do not hesitate to contact me.
Best regards
Dominika Górniewicz
On behalf of all co-authors
Military University of Technology
E-mail: dominika.gorniewicz@wat.edu.pl
Reviewer’s Comments:
Reviewer #3:
- I am glad to see that the authors took the comments seriously and attempted to make corrections in the manuscript. I see that it has improved in many areas, but I feel they should address the following issues before the paper can be accepted.
The second author of this paper is from GeniCore (https://genicore.eu/), a company that sells Spark Plasma Sintering equipment, materials (including high-entropy alloys), and also provides related services. The authors stated that they did not have any conflicts of interest. This affiliation appears to be a conflict of interest which should have been disclosed in the “Conflict of Interest” statement. Authors should explicitly declare this in their conflict of interest statement.
Response: Genicore does not deal commercially with high entropy alloys. This company works for the client by performing the commissioned task. In our case, Genicore only performs sintering using the upgraded field-assisted sintering technology (U-FAST) for the powder supplied by Military University of Technology (MUT). This company does not use the results obtained for MUT for other purposes.
The co-author of the article has knowledge and experience in the field of high-entropy alloys developed at MUT. On the other hand, work at Genicore enables additional tests to be carried out for MUT on the basis of a signed agreement between these research units. Number of the concluded contract: KJ/332/WAT/WTC/2021. Mutual cooperation is based on carrying out tests without using them for other purposes. Additionally, Genicore operates under an NDA agreement.
- In response to my comment on Fig. 1, it appears that the authors did not take it seriously. This figure adds no value to their work. They should recognize that their approach is flawed. You obtained information about the number of publications for single-phase and multiphase HEAs from the title of papers, and as a result, you may have underestimated the number of publications with multiphase, for example, the first papers on HEAs by both Cantor and Yeh included alloys with multiple phases, which cannot be counted simply by looking at the titles of their papers. The reason that titles on HEAs appear to be more focused on single-phase HEAs is because single-phase HEAs are difficult to obtain and multiphase alloys are more likely to form (again, see Cantor's first paper). However, you will not be able to obtain that information by searching the database. The authors must remove this figure and follow my advice from the previous report.
Response: We agree that this is not a review article and such graph appear in many other articles. Additionally, we agree that the title itself does not indicate what type of high entropy alloy is being described. Therefore, as suggested by the reviewer, the authors removed the plot in Figure 1, which showed the number of studies on multiphase high-entropy alloys, including alloys with Laves phases and single phase high entropy alloys based on Science Direct database. As a result, the numbering of figures has changed and Figure 1 is now replaced by another graph.

Round 3
Reviewer 2 Report
All of my concerns have been addressed appropriately.
This manuscript is a resubmission of an earlier submission. The following is a list of the peer review reports and author responses from that submission.
Round 1
Reviewer 1 Report
The Bo–Md diagram of metals is used to assess preliminarily the phase structure of high entropy alloys, and the assessment is validated by experimental characterizations in the designed TiCoCrFeMn alloy after annealing heat treatments. The manuscript provides an example for using the empirical Bo–Md diagram to the phase and composition design of multicomponent alloys, and should be interesting to the audience at Materials. The manuscript could be published after constructively address the following concerns:
- While the concept using ‘Bo-Md’ diagram for alloy design is highlighted in the title. It seems the method has been proposed and applied to the design of bcc high entropy alloys, as Ref.35 the authors cited. The novelty of using the method in the present alloy system should be clarify in the manuscript.
- When the authors comment on ‘classic design approach’ to motivate the new alloy design method. It is appreciative to briefly discuss the limit of these ‘classic design approaches’ in the description of ‘the multiphase alloys with high-entropic features’.
- The discussion related to the identification of Lave phase in the first paragraph on page 8 should be amended with convinced evidences, for instance, the XRD measurements presented in the later part.
- It is confused to conclude that “the resulting solid solution was formed on the basis of the manganese lattice” from the similar lattice parameter of Mn and the alloy.
- According to the XRD results showing Fig.9, the Lave phase didappers when the annealing time reaches to 1000h. It is appreciative that the authors can add some discussion about the reason behind. In addition, it seems that there are extra diffraction peaks belonging to the phase except bcc and hcp when the annealing time is at 10 and 20 h. The patterns should be identified with necessary discussion.
Reviewer 2 Report
Authors send a manuscript on the preparation of parts of TiCoCrFeMn multicomponent HEA alloy by mechanical alloying of five different elements and a subsequent SPS consolidation. A final heat treatment at 1000 ºC for up to 1000 h was carried out. Microstructural analysis by SEM and XRD complete the study. The studied composition was changed from a more typical composition by replacing Ni for Ti. Regarding manufacturing, and microstructural study the manuscript does not bring any new light on the study of HEA, showing expected results on these kind of alloys. The manuscript should be corrected in several points to be clearer for readers.
The main innovation of this manuscript is a new microstructural prediction method, based on the covalent bond strength and the energy level of the d orbital between the constituting elements. At the same time, this is the main concern about the content, being the method claimed as a novel design concept of HEA, but being checked just with one composition.
According to the actual content I consider the manuscript has to be rejected, and authors should be encouraged to present a similar paper, with the prediction method being checked on different compositions. It would be desirable the selected compositions leading to different microstructures, which should be accurately predicted by the claimed method.
On the other hand, points to be clarified in the manuscript are the following:
- Line 83. Include ref. for the Cantor alloy.
- Line 108 and others. Add some information in the introduction on the C14 structure, the HZ lattice, etc.
- Line 184-185 and others. Three different phases are initially described, referred as light, grey and bright. Then, in lines 194-195 bright regions seem to refer to the previous light and bright phases. The designation is not clear. You should maybe use an easier name for these phases from the first time they are mentioned in the text, and then maintain this designation throughout the manuscript.
- Line 211. The three different phases, and how they transform should be identified in the different micrographs of Figure 3.
- Line 211. I would recommend to change Fig 3e to the same magnification as others for a better comparison.
Line 212. The two BCC phases in Fig 4 show a different composition. They should be identified in the figure itself with the previously described phases.
Reviewer 3 Report
Submitted manuscript entitled "Novel design concept of high – entropy alloys with multiphase structures" is focused on characterization of microstructure of TiCoCrFeMn alloy produced by the powder metallurgy method. The authors constructed Bo-Md diagram of studied alloy. The TiCoCrFeMn alloy was then annealed and the authors focused on following changes in the chemical composition and the impact of these changes in the parameters of the crystal structures of the identified phases in the Bo-Md diagram model. The way how is the research story delivered and how are the scientific conclusions supported by evidence and proper discussion with existing literature should be significantly improved. Several remarks and suggestions which should be addressed before the manuscript can be considered for publication are summarized in what follows:
- The English language has to be improved to maintain standards requested by the Materials journal. Please attend to this. There are type-setting errors and grammar mistakes.
- Figure 1 – Caption to Figure 1 is incorrect, line plots corresponding to HEA is green in the graph, not blue as mentioned in captions. “HEA alloys with the participation of the Laves phase” is blue in the graph, not green as mentioned in the captions.
- Page 3 – At the end of Introduction section, a paragraph should be added/formulated stating clearly what is the main goal of the current work presented in the manuscript. The main goal of current work following motivation is not clear currently.
- Page 4 – Experimental section – “…, where the chemical composition of EDS and X-ray microanalyses were performed using Electron Backscatter Diffraction (EBSD) detector.” Please clarify this statement. Chemical composition of materials is usually determined using energy-dispersive X-ray spectroscopy (EDS) detector, NOT by EBSD.
- Page 4 – Please provide evidence or references to evidence with the values of time of degradation for Ti, Cr, Mn, Fe and Co.
- Page 4 - Please provide reference for statement, that “Based on British standards, only elements of low-level waste, that will degrade within 100 years could be used in the construction of nuclear power components.”
- Page 5 – Please provide clear and detailed explanation for the statement “Such location suggests that the final phase structure will consist of the BCC solid solution and the Laves C14 phase with the HCP lattice”. How is this concluded from the diagram in Figure 2? Based on what facts is this statement correct? Please clarify.
- Page 5 – What authors mean by “two-phase structure consisting of light and grey areas with a grain size below 1 um (Figure 3a, b)“? Authors need to explain what type of experimental techniques they used to study the microstructure of the material. There is not such a term in material science as “dark and bright phase”. The authors can speak about brighter or darker contrast in reference to the experimental technique used and the way, how the contrast is formed. For example, in BSE, contrast can be sensitive to chemical composition of different phases but also it can be related to different crystal orientations of different grains in polycrystalline material. The authors did not provide enough information about how the characterization was done and what data and what conclusions can be made based on these data. In Figure 3, the authors vaguely reference to images (a, c, e) as “EBSD detector” – however, it is absolutely not clear how are the data from the EBSD plotted? It seems like “image quality map”, which is however not appropriate way. The phase map or crystal orientation map is more accepted visualization of EBSD data.
- Page 7 – How was the “microanalysis” in Figure 4 done? What technique and what parameters were used to obtain Wt.% and At.% showed in the Figure 4? What software was used to quantify collected raw data so Wt.% and At.% were obtained? What is the preciseness of the method (mean deviation)? Please clarify.
- Page 7 – The Figure 5 has to be completely re-done. From the Figure 5, it is absolutely not clear for the reader, that the analysis of the areas of “large bright particles” indicate their BCC lattice structure with a parameter (what parameter? Lattice parameter?) similar to the manganese lattice. Please, clarify and provide clear evidence for your statements.
- Page 8 – what technique was used to obtain data about chemical content in Table 2? Please clarify in detail.
- Page 8 – The notation (lb, sb and sg) used in Table 2 and throughout the text is very confusing. Characterization of different phases should be done more “scientifically”. For example, the area of interest investigated in terms of SEM-BSE should be mapped also by SEM-EBSD and SEM-EDS techniques. These maps should be plotted in addition and next to the SEM-BSE image. Then, the characterization of three distinctive areas and their identification should be done on the objective scientifically measurable values. The areas currently referenced in the manuscript as “bright” or “grey” have differences in SEM-BSE contrast based on scientifically determinable structural characteristics such as crystal lattice difference, chemical composition, etc.
